# Beyond safety drivers: Applying air traffic control principles to support the deployment of driverless vehicles

Robert C. Hampshire[1,2☯‡]*, Shan Bao[2,3‡], Walter S. Lasecki[4‡], Andrew Daw[5☯‡], Jamol Pender[5,6☯‡]

**1** Gerald R. Ford School of Public Policy, University of Michigan, Ann Arbor, MI, United States of America, **2** Transportation Research Institute, University of Michigan, Ann Arbor, MI, United States of America, **3** Industrial and Manufacturing Systems Engineering, University of Michigan–Dearborn, Dearborn, MI, United States of America, **4** Computer Science and Engineering, University of Michigan, Ann Arbor, MI, United States of America, **5** Operations Research and Information Engineering, Cornell University, Ithaca, NY, United States of America, **6** Center for Applied Mathematics, Cornell University, Ithaca, NY, United States of America

☯ These authors contributed equally to this work.
‡ These authors contributed equally to the research conceptualization and investigation and to editing the manuscript.
* hamp@umich.edu

**Data Availability Statement:** The paper uses publicly available US survey data: https://nhts.ornl.gov/.

## Abstract

By adopting and extending lessons from the air traffic control system, we argue that a nationwide remote monitoring system for driverless vehicles could increase safety dramatically, speed these vehicles' deployment, and provide employment. It is becoming clear that fully driverless vehicles will not be able to handle "edge" cases in the near future, suggesting that new methods are needed to monitor remotely driverless vehicles' safe deployment. While the remote operations concept is not new, a super-human driver is needed to handle sudden, critical events. We envision that the remote operators do not directly drive the vehicles, but provide input on high level tasks such as path-planning, object detection and classification. This can be achieved via input from multiple individuals, coordinated around a task at a moment's notice. Assuming a 10% penetration rate of driverless vehicles, we show that one remote driver can replace 14,840 human drivers. A comprehensive nationwide interoperability standard and procedure should be established for the remote monitoring and operation of driverless vehicles. The resulting system has potential to be an order of magnitude safer than today's ground transportation system. We articulate a research and policy roadmap to launch this nationwide system. Additionally, this hybrid human–AI system introduces a new job category, likely a source of employment nationwide.

## Introduction

By adopting and extending lessons from the air traffic control system, a nationwide remote monitoring system for driverless vehicles could dramatically increase safety, speed deployment

**Funding:** This work is sponsored by the U.S. Department of Transportation Center for Connected and Automated Vehicles (CCAT), based at the University of Michigan's Transportation Research Institute.

**Competing interests:** The authors have declared that no competing interests exist.

of these vehicles, and provide a source of employment in this nascent industry. In 2018, California became the first state in America to permit driverless vehicle testing on public roads. Sensibly, state law [1] requires a permit holder to certify that:

"There is a communication link between the vehicle and the remote driver to provide information on the vehicle's location and status and allow two-way communication between the remote driver and any passengers if the vehicle experiences any failures that would endanger the safety of the vehicle's passengers or other road users, or otherwise prevent the vehicle from functioning as intended, while operating without a driver."

Furthermore, the permit holder must provide to remote drivers:

"Instruction on the automated driving system technology being tested, including how to respond to emergency situations and hazardous driving scenarios that could be experienced by the vehicle or the vehicle's occupants."

Remote drivers play an important role as "traffic control" to monitor, plan, and possibly actively support the safety of driverless vehicle passengers and other road users. While we acknowledge possible security vulnerabilities of this approach, we call on the research and technical communities to develop secure means to enable remote monitoring and operations. Regrettably, recent legislation in Nebraska [2] and pending legislation in Alabama [3] (although the proposed Alabama law explicitly allows for teleoperations systems) and Missouri [4] do not require a remote driver for testing on those states' public roads, according to the National Conference of State Legislatures [5]. "Edge cases" are widely recognized to exist, for which online back-up human assistance is needed to guarantee safety [6, 7]. A teleoperations system might have prevented the first pedestrian fatality involving a self-driving vehicle.

On Sunday, March 18, 2018, at 10pm a self-driving vehicle operated by Uber struck and killed 49-year-old Elaine Herzberg in Tempe, Arizona, as she walked across a lane of traffic. A safety driver in the car, not attending to the road, was alerted of the pending crash too late [8]. The vehicle's light detection and ranging (LIDAR) system first sensed the pedestrian 6 seconds before the fatal crash. At that time, the vehicle was travelling 43 miles per hour, approximately 378 feet from the pedestrian. Only 1.3 seconds before impact, the vehicle engaged emergency braking. What was happening during the intervening 4.7 seconds? According to the official National Transportation Safety Board (NTSB) report, the vehicle object detection software could not confidently identify the observation during that time interval.

We argue that a teleoperations system for a fleet of driverless vehicles would efficiently use those 4.7 seconds to improve road safety and reduce crash severity. Recent insights into human-assisted artificial intelligence (AI) systems establish the feasibility of response times less than 0.3 seconds (roughly equivalent to the average human reaction time, but more stable and less subject to distraction) by combining reinforcement learning with crowd feedback [9]. The system we envisage will leverage a call center staffed by skilled remote human drivers who monitor and assist autonomous vehicles' driving tasks. This includes both critical (emergency braking, lane departure, etc.) and non-critical tasks (passenger pick-up and drop-off, navigating road construction, etc.). The teleoperations system objective is to prevent such critical crashes as well as to handle noncritical scenarios.

Remote operator and monitoring systems are already in common use. Fig 1 shows a typical setup. We envision that remote operators will provide control inputs to the vehicles when a request is made to the teleoperations call center. A example of a potential remote operator is given in Fig 1. On the left of Fig 1, we have a visual of a remote agent being trained by a

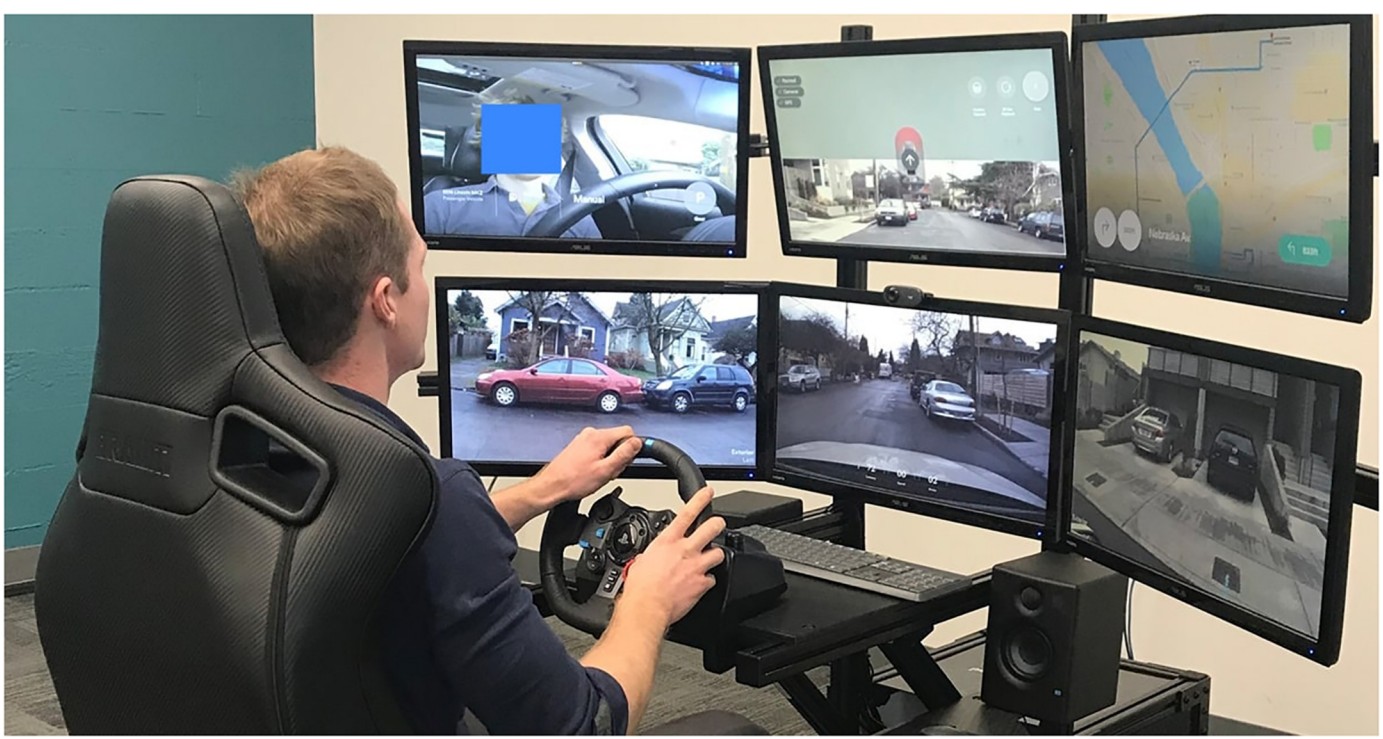

**Fig 1. Teleoperations platform by the startup Designated Driver (https://designateddriver.ai/).**

company Designated Driver. Our goal is understand how to staff and operate a teleoperations system with these remote operators performing driving assistance tasks.

## Remote drivers as AI-coordinated groups

Automated vehicles promise a host of societal benefits, including dramatically improved safety, increased accessibility, greater productivity, and higher quality of life. In order to deliver on these promises, the vehicles must be able to operate and reason over a nearly infinite number of known and unknown potential conflict situations. By adopting lessons and experiences from the air traffic control system, we argue that using humans to supervise driverless vehicles is 1) technically feasible, 2) necessary to achieve safety goals, and 3) an available source of employment [10].

Fully autonomous (Level 5) vehicle control is a relatively distant goal, as it requires AI to both understand scenarios involving people and other objects in the environment and know how to respond. Current autonomous vehicles (AVs) can drive quite well in typical (frequently encountered) settings but fail in exceptional cases. Worse, these exceptional cases are often the most dangerous and may arise suddenly, leaving human drivers with only a couple of seconds, at most, to react—precisely a setting in which people can be expected to perform worst. Compared to precautionary takeovers, these sudden scenarios already comprise most of the disengagements that GM Cruise reported in 2018, accounting for 53.5% of autonomous driving interruptions [11]. As self-driving capabilities improve, non-time-critical disengagements should become increasingly rare. For uncommon situations that are not time-critical (e.g., when the vehicle is stopped), remote human drivers can navigate and return the AV to a setting from which it can resume autonomous control. However, in critical settings where rapid response is needed (e.g., sudden events that could result in collision), asking a remote human

driver to step in is not viable due to the latency period required for a driver to perceive/understand the context of a scenario and react, as well as the network latency involved in transmitting video and responses between safety drivers and vehicles.

Our team's current work explores how AI-coordinated groups of remote drivers might best attain superhuman collective performance, overcoming previously insurmountable barriers of human and network latency. By leveraging AVs' "understanding" of the world (e.g., state space representation, transition model), human effort/insight can be guided toward reachable future states of the world. This allows us to simulate potential situations mere seconds or even fractions of a second before they occur, and cache responses indicating to the AV how a human would respond locally in such a situation—the result being an ability to leverage human responses in milliseconds rather than seconds, opening a whole new frontier for possible applications to critical, life-saving scenarios. See [9] for more details on the foundational techniques underlying our approach.

## Results: How many remote drivers are needed?

A naive approach to staffing remote drivers would dedicate one to each vehicle in the fleet. Less extreme could be one agent per active vehicle. However, even this is quite extreme: that all vehicles would require simultaneous assistance is highly unlikely. Thus, we now demonstrate a staffing approach that is significantly more efficient without taking on significant risk. Using the 2017 National Household travel survey (NHTS) [12] and 2018 self-reported AV disengagements in California [11], we estimate the number of remote drivers needed to staff the teleoperations control system. The lowest reported disengagement rate is 1 disengagement per 11,000 miles. In 2017, 2.1 trillion passenger miles were driven nationwide. If we assume a 10% penetration rate of driverless vehicles in the fleet of all vehicles traversing the nation, 6.25 million disengagements might occur during peak hours each year, or roughly 17,000 disengagements during the busiest hour each day. This is calculated by aggregating the average miles driven each hour across the country as provided in [12], and then converting this number of miles to the number of disengagements via the aforementioned disengagement rate. Our calculations are described further in the Materials and Methods section.

Table 1 shows both the annual passenger miles driven and the estimated number of remote drivers needed per hour to handle disengagements. The results suggest that one agent is needed for approximately every 200 million miles driven annually (Fig 2). The average

**Table 1. Estimated number of remote drivers needed during peak travel time under three arrival types.**

| Rank | Metropolitan area | Total annual miles driven in area (Millions) | Number of remote drivers needed (Standard arrivals) | Number of remote drivers needed (Bursty arrivals) | Number of remote drivers needed (Highly bursty arrivals) |
|---|---|---|---|---|---|
| 1 | New York, NY | 93,512 | 103 | 111 | 122 |
| 2 | Los Angeles, CA | 71,791 | 83 | 89 | 100 |
| 3 | Dallas, TX | 50,231 | 62 | 67 | 76 |
| 4 | Chicago, IL | 49,348 | 61 | 66 | 75 |
| 5 | Atlanta, GA | 42,547 | 54 | 59 | 67 |
| 6 | Houston, TX | 42,431 | 54 | 59 | 67 |
| 7 | Washington, DC | 41,199 | 53 | 58 | 66 |
| 8 | Minneapolis, MN | 34,540 | 46 | 51 | 58 |
| 9 | Philadelphia, PA | 32,781 | 44 | 49 | 56 |
| 10 | Phoenix, AZ | 31,408 | 43 | 47 | 54 |

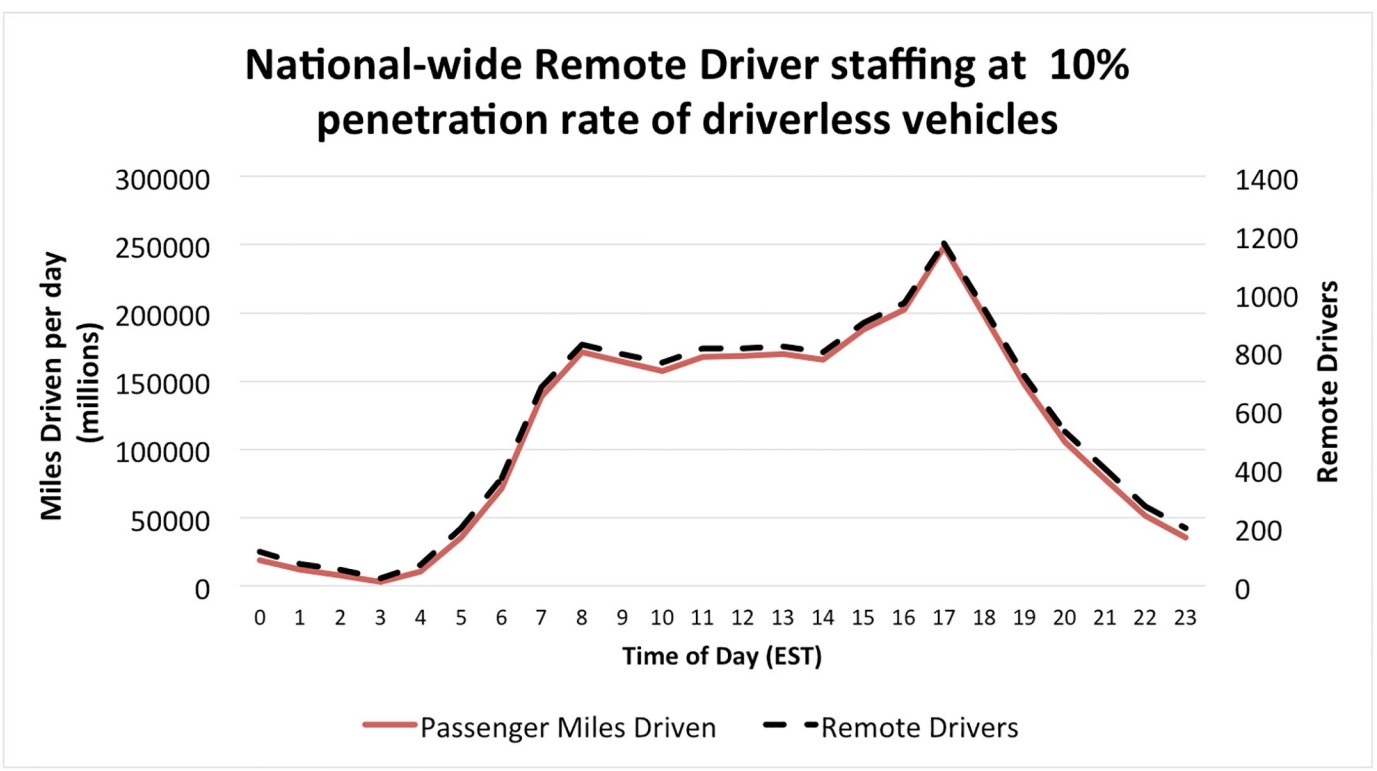

**Fig 2. Estimated number of remote drivers needed across the United States.** Calculated by time of day in the Erlang-B model. All times normalized to Eastern Standard Time.

American drives 13,476 miles each year. Thus one remote driver could replace approximately 14,840 drivers if each disengagement task is assigned to a remote driver, representing potentially massive savings in human time and attention. Details of this calculation are shown in the Materials and Methods section of this paper. Because initial deployments may opt for assigning multiple remote drivers to each disengagement, these estimates for the number of remote drivers are a lower bound. Furthermore, several remote drivers may be assigned the same task to ensure robust, accurate decisions. For these reasons, remote support centers for AVs may actually employ quite a large workforce.

## Discussion: Building blocks of a teleoperations system

An industry of AV software and hardware makers exists, as well as several startups developing teleoperations systems for driverless vehicles. While remote operations itself is not a new concept, what is needed is a super-human driver for sudden, critical events. This can be achieved via input from multiple individuals, coordinated around a task at a moment's notice. We now detail three key building blocks that are required to achieve this vision: human-assisted AI, the human element, and system-level organization.

### Human-assisted AI

With the rise of artificial intelligence as a service, human-backed algorithms at scale have become the norm rather than the exception for intelligent systems. Google, Facebook, Apple (Siri), Samsung, Bloomberg, and countless other organizations use large groups of human

annotators and checkers to ensure their intelligent services' quality and reliability. However, while reliability and accuracy are important in all these settings, none of the prior methods has leveraged low-latency, real-time systems to provide input faster than any one person alone could.

Here's how it can work—*within five seconds or less*:

- Software in the autonomous vehicle would analyze real-time vehicle data and electronically estimate the likelihood of "disengagement"—due to a situation in which the car's automated systems might need human help—10–30 seconds in the future.

- If the likelihood exceeds a pre-set threshold, the system contacts a remotely located control center, sending data from the car. One or more remote drivers are assigned to resolve the pending disengagement.

- The control center's system analyzes the car's data, generates several possible scenarios, and provides them to several human supervisors situated in driving simulators.

- The remote drivers respond to the simulations and their responses are sent back to the vehicle.

- The vehicle now has a library of human-generated responses that it can choose from instantaneously, based on information from on-board sensors.

Previous work establishes the feasibility of this approach in low-latency environments. Responses in < 0.3 s (roughly equivalent to the human reaction time, but more stable and less subject to distraction) by combining reinforcement learning with crowd feedback [9]. More work is needed to formally model collective input mediation strategies that can optimize for either input reliability or low latency [13].

The proposed system asks groups of remote drivers to help concurrently with a given monitored or control task in as little as $\sim 350$ ms of a need's arising. With video-based remote control latencies as low as 100 ms, total latency for control can be under 0.5 s. Thus, in any environment where we can predict possible outcomes 0.5 s in the future, "instant" responses become possible. This scope of settings is far larger than those we can observe prior to deployment. Prior work learned how to effectively interleave and combine groups' input over short time spans [9]. More work is needed to modify these approaches to workloads with short, sudden bursts of requests.

To improve response speed, methods are needed to directly leverage the AV's ability to understand possibilities that may arise in real settings (even when the system does not know how to respond to a possible setting) to pre-fetch possible configurations of the world. Using these future states, remote drivers can (in parallel) provide feedback before a system needs to know what action to take. What makes this possible is the speed of existing real-time staffing approaches. While 0.5 s may be a relatively slow response time for an engaged driver to respond to an event (usually accomplished within 200-300 ms), an ability to respond this quickly means that we need only pre-fetch future states of the world.

Recent work has shown that just-in-time (JIT) training can result in an average response time below 3.5 ms, reducing latency by three orders of magnitude [9]. Further, the collective response is more likely to be correct than a single person's (i.e., a local driver's) response [14]. The remaining challenge is to scale up from laboratory settings (simple, fully controlled problems) to real-world settings with massive state spaces. To improve scalability, research is needed on workload, arousal levels, and task routing optimization to utilize remote drivers' time and attention effectively and efficiently. This work must cost-effectively optimize the next selected set of states, as well as worker availability and response speed to reduce the horizon

needed to train a system for an event. Improved machine learning algorithms are also required to integrate task- and scenario-specific knowledge, to form teams of remote drivers who have performed well in similar settings (e.g., navigating a front-wheel drive car in the rain with minor driver distraction). More generally, this building block is concerned with group efficiency, collective human performance on critical tasks, and ad hoc team formation to do highly skilled work.

## The human element

Air traffic controllers are well known to have one of the most stressful jobs in the world [15]. The job is both cognitively and physically challenging, as they are responsible for maintaining and managing all incoming and outgoing air traffic, which requires highly sustained concentration and decision making. We expect the role of remote driver will be similarly demanding and challenging to deal with all routine and unexpected situations. In general, under stress, human cognitive and perceptual motor performance are both impaired [16]. Human memory, especially the encoding and maintenance processes, are very sensitive to stress effects, due to reduced resource capacity [16]. Negative effects of stress on perceptual and psycho-motor tasks have also been reported consistently. For example, Scerbo [17] examined human sustainable attentional processes and suggested that vigilance under stress can lead to decremental motor performance accuracy and increased response time. Although they constitute distinct processes and outcomes, both human judgment and decision making under generally stressful conditions tend to become less flexible, with fewer alternatives considered [18, 19].

While much research has been done on air traffic controllers, more will be needed to understand how remote drivers perform on monitoring and operating tasks. Performance measures should include evaluating accuracy, service time, cognitive load, and fatigue resulting from processing service requests. Understanding interactions of the above factors within a single person or team is critical to the safety performance of the teleoperations system. The findings of the human factors research should inform a licensing standard for remote drivers. The state of California statute requires the testing permit holder to document and certify that the remote drivers have adequate training and education. Such an approach creates standards of safety and certification needed to create a professional job category for remote drivers. Further research is needed to estimate the number of available workers with requisite cognitive and emotional skills, and judgement to become remote drivers.

## System operations and industrial organization

This teleoperations system might have several possible operating models. In one, private companies who own or operate driverless vehicles would also own and operate remote assistance centers. This model is similar to the current GM OnStar system, in which only GM-equipped vehicles can access OnStar. This approach would allow the industry to compete on safety. The teleoperations system would be a feature that users or fleet purchasers can choose much like adaptive cruise control (ACC). However, this approach would likely lead to balkanized teleoperations systems that do not talk to each other. Standards groups like the Society of Automotive Engineers (SAE) or the International Organization for Standardization (ISO) would need to set standards to promote interoperability and communication between teleoperations systems. Additionally, this approach would require employing more drivers, due to a scale smaller than a centralized system.

Another model resembles the air traffic control system operated by the Federal Aviation Administration (FAA). Vehicle support tasks are split between workers at various local, regional, and national centers. As a vehicle moves between various locations, oversight is

handed off between centers. A private firm under contract with the federal government could operate this system. If an example is wanted, a private nonprofit runs the nationwide air traffic control system in Canada [20]. This type of arrangement, under which oversight is handed off between different regions, can be modeled in a way that is similar to wireless communication networks [21, 22]. In this type of model, one separates the state space into two parts: One dimension consists of those drivers on the road who need no assistance; the other dimension consists of drivers who do assistance. This type of model has also been used in the context of healthcare, to model emergency-room patients in critical and stable conditions [23].

## Materials and methods

We base the staffing estimates on 2017 nationwide passenger vehicle driving statistics and the disengagements generated from daily passenger travel in the United States. The National Household Travel Survey (NHTS) provides the annual miles driven for each hour of the day. We aggregate all the demand by shifting times to Eastern Standard Time (EST), and focus this analysis on the 52 metropolitan statistical areas (MSA) with more than 1 million people. We develop two queueing models to estimate the remote driver staffing levels [24]. One model is a blocking model; the other is a delay model. In the blocking model, service requests to the remote driver system are denied and immediately leave the system if all remote drivers are servicing other requests. In the delay model, service requests wait their turn in a queue if all remote drivers are occupied. All of the numerical results are based on a scenario with 10 percent penetration rate of driverless vehicles, 1 disengagement per 11,000 miles, and a 10-minute average service time per disengagement.

### Blocking model perspective

We model the number of remote drivers needed as an Erlang-loss queueing system. We assume, for the sake of simplicity, that requests for service to our dynamic queueing system is driven by a stationary Poisson process with rate λ. This allows us to perform staffing calculations for peak hour demand. In the subsequent subsection we discuss how to extend these computations to time-varying staffing settings, but for now let us consider the peak hour.

$$B_C(q) = \frac{\frac{q^C}{C!}}{\sum_{i=0}^{C} \frac{q^i}{i!}} = \frac{P(Q_\infty = C)}{P(Q_\infty \leq C)}. \tag{1}$$

In this formula, the offered load $q$ is defined as the arrival rate λ of requests for assistance divided by the rate of service μ. Moreover, the blocking probability can be calculated as the conditional probability of an infinite server queue's being in state $C$ given that the queue length never exceeds $C$. Using this formula, one observes that in order to satisfy $1 - \epsilon$ fraction of all service requests immediately, a minimum number of drivers ($C$) is needed such that $B_C(q) \leq \epsilon$. Finding such a $C$ normally would be difficult numerically because of the factorial in the blocking probability expression. A method exists, however, to solve for the number of drivers recursively. Known as the *Erlang-B recursion*, an expression for this method is shown below:

$$B_C(q) = \frac{qB_{C-1}(q)}{C + qB_{C-1}(q)}. \tag{2}$$

In addition to the recursion, one can derive simple, accurate approximations for the number of drivers needed to satisfy $1 - \epsilon$ of all service requests immediately. Also of note: the Erlang-B formula is valid for the *M/G/C/C* queue, where the service time distribution is general, since the Erlang-B formula is beautifully insensitive to the distribution of this random variable.

However, insensitivity is not true for the vehicle arrival process. Thus, for the $G/G/C/C$ queue, the Erlang-B formula needs not hold, so approximations are needed to compute the number of drivers necessary to immediately satisfy percent of all service requests. When inter-arrival and service time distributions are exponential, an approximation of the minimum number of required drivers [25, 26] equals

$$C = q + x\sqrt{q}, \qquad (3)$$

in which $x$ satisfies the following inequality

$$\frac{1}{\sqrt{q}} \frac{\phi(x)}{\Phi(x)} = \varphi(x) \leq \epsilon. \qquad (4)$$

This formula relies on the probability density function (PDF), $\phi(\cdot)$, and the cumulative distribution function (CDF), $\Phi(\cdot)$, of a standard normal random variable, which can be quickly calculated. To derive this formula, we start with the Erlang-B formula in terms of the infinite server queue. Note that when the arrival rate is large or demand for servers is sufficient, the infinite server queue can be approximated by a Gaussian distribution. The distribution of the infinite server queue in steady state is Poisson, satisfying the condition that all of its cumulant moment equal the offered load [27]. In particular, the mean equals the variance, which implies that $Q_\infty \approx N(q, q)$. Using this Gaussian approximation, we can derive an approximate formula for the number of drivers needed to satisfy the blocking probability requirements:

$$B_C(q) = \frac{P(Q_\infty = C)}{P(Q_\infty \leq C)} \approx \frac{P(N(q, q) = C)}{P(N(q, q) \leq C)} = \frac{\phi\left(\frac{C-q}{\sqrt{q}}\right)}{\Phi\left(\frac{C-q}{\sqrt{q}}\right)\sqrt{q}} = \varphi\left(\frac{C-q}{\sqrt{q}}\right) \leq \epsilon. \qquad (5)$$

Now, by inverting the function $\varphi(x)$, we see that the number of drivers needed to satisfy our blocking probability is approximately equal to:

$$C \approx q + \varphi^{-1}(\epsilon)\sqrt{q}. \qquad (6)$$

When the mean does not equal the variance, $Q_\infty \approx N(q, v)$, which yields a slightly different formula for the approximate number of drivers needed to satisfy the blocking probabilities. Following the same approach as above, we find:

$$B_C(q) = \frac{P(Q_\infty = C)}{P(Q_\infty \leq C)} \approx \frac{P(N(q, v) = C)}{P(N(q, v) \leq C)} = \frac{\phi\left(\frac{C-q}{\sqrt{v}}\right)}{\Phi\left(\frac{C-q}{\sqrt{v}}\right)\sqrt{q}} = \varphi_v\left(\frac{C-q}{\sqrt{q}}\right) \leq \epsilon. \qquad (7)$$

By inverting this new function $\varphi(x)$, we find that the number of drivers needed to satisfy our blocking probability is now approximately equal to:

$$C \approx q + \varphi_v^{-1}(\epsilon)\sqrt{v}. \qquad (8)$$

This expression is equivalent to the Hayward approximation [26, 28, 29, 30]. Why does the Erlang-B formula need to be so modified? For one thing, inter-arrival times and service times are not expected to be independent and identically distributed. Imagine a fallen tree blocking the road: The first vehicle whose driver observes the downed tree might disengage or need assistance. However, vehicles close behind might also require assistance, for the same reason. Thus, arrivals might tend to cluster during events and require similar service times since they reference the same type of disengagement.

## Extension to time-varying calculations

In this section, we will discuss the generalization to non-stationary arrival rates. In doing so, we derive closed form formulas for mean queue length of the $M_t/G/\infty$ queueing model. These results are not new as they were derived in [31, 32] for the time varying infinite server queue. Eick et al. use the properties of the Poisson arrival process and use Poisson random measure arguments to show that the $M_t/G/\infty$ queue $Q_\infty(t)$, has a Poisson distribution with time varying mean $q(t)$ that is known [32]. The infinite server queue is an important model to study despite it having an infinite number of servers since it represents the queueing process as if there were an unlimited amount of resources to satisfy the demand process. In fact, [32] show that $q(t)$ has the following probabilistic representation

$$q(t) = E[Q_\infty(t)] = \int_{-\infty}^{t} \bar{G}(t-u)\lambda(u)\mathrm{d}u = E\left[\int_{t-S}^{t} \lambda(u)\mathrm{d}u\right] = E[\lambda(t-S_e)]E[S], \qquad (9)$$

where $\lambda(u)$ is the time varying arrival rate at time $u$, $S$ represents a service time with distribution $G$, $\bar{G}(x) = 1 - G(x) = \mathbb{P}(S > x)$, and $S_e$ is a random variable with distribution that follows the stationary excess of residual-lifetime CDF $G_e$, defined by

$$G_e(t) \quad \equiv \quad \mathrm{P}(S_e < t) = \frac{1}{E[S]}\int_0^t \bar{G}(u)\mathrm{d}u = \frac{1}{E[S]}\int_0^t \mathrm{P}(S > u)\mathrm{d}u, \quad t \geq 0. \qquad (10)$$

When the service time distribution is exponential, we know that the mean queue length, $q(t)$, solves the ordinary differential equation

$$\frac{\mathrm{d}}{\mathrm{d}t}q(t) = \lambda(t) - \mu \cdot q(t). \qquad (11)$$

In fact, since the differential equation is linear non-homogeneous ordinary differential equation, the solution is given by

$$q(t) = q_0 e^{-\mu t} + e^{-\mu t}\int_0^t \lambda(s)e^{\mu s}\mathrm{d}s \qquad (12)$$

Moreover, from the standard theory of infinite server queues, the distribution of the queue length process is Poisson with mean $q(t)$ when initialized with a Poisson number customers or initialized at zero.

Recent work by [33] and [34] uses the infinite server queue to develop staffing algorithms for multi-server queues. Like in the stationary case, the number of servers needed to achieve a blocking probability of $\epsilon$ is given by the time varying function

$$C(t) \approx q(t) + \varphi^{-1}(\epsilon)\sqrt{q(t)}. \qquad (13)$$

From [35] and [34], it has been shown that Eq 13 can be used to stabilize the blocking and delay probabilities regardless of the arrival rate and in some cases the arrival and service rate distributions. We find it also important to make the comment that it clearly suffices to analyze a stationary model for staffing purposes if the arrival rate is given by $\lambda = \sup_{0 \leq s \leq T} \lambda(s)$. One can observe the similarity between Eqs 13 and 6. Using this arrival rate will certainly produce an overestimate of the number of servers needed, however, it will most definitely satisfy the probabilistic constraints. Thus, this both justifies our peak hour analysis and demonstrates how it can be extended to time-varying settings.

## Delay model perspective

In addition to the Erlang-B model, we can also use the Erlang-C model for situations in which vehicles wait for an agent to provide service. In this section, we assume the inter-arrival and service time distributions are exponential, with rates $\lambda$ and $\mu$, respectively. Under these assumptions, the probability of delay is given by the following expression:

$$\mathrm{P}(\text{Wait} > 0) = \mathrm{P}(Q \geq C) = \frac{\frac{q^C}{C!\left(1-\frac{q}{C}\right)}}{\frac{q^C}{C!\left(1-\frac{q}{C}\right)} + \sum_{k=0}^{C-1} \frac{q^k}{k!}} = \frac{B_C(q)}{1 - \frac{q}{C}(1 - B_C(q))}. \tag{14}$$

We can exploit our knowledge of the $M/M/C$ queue by using its conditional waiting time distribution, which we know to be exponential. Thus, we can also know that:

$$\mathrm{P}(\text{Wait} > w \mid \text{Wait} > 0) = \mathrm{P}(\text{Wait} > x \mid Q \geq C) = e^{-(C\mu - \lambda)} \leq \epsilon. \tag{15}$$

Solving for the number of drivers to satisfy the excessive wait probability yields:

$$C = q - \frac{\log(\epsilon)}{\mu w} \tag{16}$$

Similarly, we can use the expected delay formula for the $M/M/C$ queue to derive the number of drivers that would satisfy a bound on the expected delay for a driver experiencing disengagement:

$$\mathrm{E}[\text{Wait} \mid \text{Wait} > 0] = w = \frac{1}{\mu C - \lambda}. \tag{17}$$

Solving for the number of drivers to satisfy the expected conditional delay yields:

$$C = q + \frac{1}{\mu w}. \tag{18}$$

## Calculating staffing levels from data

Table 2 shows both the annual passenger miles driven and the estimated number of remote drivers needed per hour to handle disengagements under different staffing models. As discussed at the beginning of this Materials and Methods section, the data in this table were sourced from the 2017 National Household Travel Survey (NHTS). The rate of one disengagement per 11,000 miles is based on 2018 California disengagement reports. Table 2 differs from Table 1 in presenting the number of remote drivers for some of the largest cities in the nation for a greater variety of staffing models. From left to right, we have the city name, annual number of miles driven, percent of miles driven nationwide, remote drivers given by the Erlang-B model (with blocking probability 0.001), remote drivers given by the probability of delay in the Erlang-C model (with delay probability 0.001), remote drivers given by the conditional mean wait of the Erlang-C model (with a target wait of 20 s), remote drivers given by the conditional excessive wait of the Erlang-C model (with only 0.1% of waiting times exceeding 60 s), and conditional Gaussian approximation using peakedness (variance to mean ratio) parameters {0.25, 0.5, 1, 2, 4}. Because the ratio of the variance to the mean serves as a measure of under- or over-dispersion, cases in which the peakedness parameter is greater than one approximate models with bursts of arrivals [36–39]. Table 2 indicates that most of the queueing models suggest staffing a similar number of remote drivers. The largest differences in number of remote

**Table 2. Estimated number of remote drivers needed in select cities nationwide during peak travel time, under five staffing models.**

| Core Based Statistical Area (CBSA) | Annual Miles Driven (mil.) | Percent National Miles Driven | Erlang B | Erlang C (delay) | Erlang C (mean wait) | Erlang C (excess wait) | Normal Approx. (z = 0.25) | Normal Approx. (z = 0.5) | Normal Approx. (z = 1) | Normal Approx. (z = 2) | Normal Approx. (z = 4) |
|---|---|---|---|---|---|---|---|---|---|---|---|
| Atlanta-Sandy Springs-Roswell | 42,547 | 2.0 | 54 | 56 | 66 | 66 | 45 | 49 | 53 | 59 | 67 |
| Austin-Round Rock | 18,664 | 0.9 | 29 | 30 | 46 | 46 | 22 | 25 | 28 | 32 | 38 |
| Baltimore-Columbia-Towson | 17,217 | 0.8 | 27 | 28 | 45 | 45 | 21 | 23 | 26 | 30 | 36 |
| Birmingham-Hoover | 9,087 | 0.4 | 18 | 18 | 38 | 38 | 13 | 14 | 17 | 20 | 24 |
| Boston-Cambridge-Newton | 25,440 | 1.2 | 36 | 38 | 52 | 52 | 29 | 32 | 35 | 40 | 47 |
| Buffalo-Cheektowaga-Niagara Falls | 5,370 | 0.3 | 13 | 13 | 35 | 35 | 9 | 10 | 12 | 14 | 18 |
| Charlotte-Concord-Gastonia | 17,968 | 0.9 | 28 | 29 | 45 | 45 | 22 | 24 | 27 | 31 | 37 |
| Chicago-Naperville-Elgin | 49,348 | 2.3 | 61 | 63 | 71 | 71 | 51 | 55 | 60 | 66 | 75 |
| Cincinnati | 17,080 | 0.8 | 27 | 28 | 45 | 45 | 21 | 23 | 26 | 30 | 36 |
| Cleveland-Elyria | 14,238 | 0.7 | 24 | 25 | 42 | 42 | 18 | 20 | 23 | 27 | 32 |
| Columbus | 15,759 | 0.7 | 26 | 27 | 44 | 44 | 20 | 22 | 25 | 29 | 34 |
| Dallas-Fort Worth-Arlington | 50,231 | 2.4 | 62 | 64 | 72 | 72 | 52 | 56 | 61 | 67 | 76 |
| Denver-Aurora-Lakewood | 18,972 | 0.9 | 29 | 30 | 46 | 46 | 23 | 25 | 28 | 33 | 38 |
| Detroit-Warren-Dearborn | 26,001 | 1.2 | 37 | 38 | 52 | 52 | 30 | 32 | 36 | 41 | 48 |
| Grand Rapids-Wyoming | 7,866 | 0.4 | 16 | 17 | 37 | 37 | 11 | 13 | 15 | 18 | 22 |
| Hartford-West Hartford-East Hartford | 7,843 | 0.4 | 16 | 17 | 37 | 37 | 11 | 13 | 15 | 18 | 22 |
| Houston-The Woodlands-Sugar Land | 42,431 | 2.0 | 54 | 56 | 66 | 66 | 45 | 48 | 53 | 59 | 67 |
| Indianapolis-Carmel-Anderson | 10,398 | 0.5 | 19 | 20 | 39 | 39 | 14 | 16 | 18 | 22 | 26 |
| Jacksonville | 8,134 | 0.4 | 17 | 17 | 37 | 37 | 12 | 13 | 15 | 19 | 23 |
| Kansas City | 10,969 | 0.5 | 20 | 21 | 40 | 40 | 15 | 17 | 19 | 22 | 27 |
| Las Vegas-Henderson-Paradise | 8,809 | 0.4 | 17 | 18 | 38 | 38 | 12 | 14 | 16 | 19 | 24 |
| Los Angeles-Long Beach-Anaheim | 71,791 | 3.4 | 83 | 86 | 90 | 90 | 72 | 76 | 82 | 89 | 100 |
| Louisville/Jefferson County | 10,274 | 0.5 | 19 | 20 | 39 | 39 | 14 | 16 | 18 | 22 | 26 |
| Memphis | 6,386 | 0.3 | 14 | 15 | 36 | 36 | 10 | 11 | 13 | 16 | 20 |
| Miami-Fort Lauderdale-West Palm Beach | 28,918 | 1.4 | 40 | 42 | 55 | 55 | 32 | 35 | 39 | 44 | 51 |
| Milwaukee-Waukesha-West Allis | 9,509 | 0.5 | 18 | 19 | 38 | 38 | 13 | 15 | 17 | 20 | 25 |

*(Continued)*

**Table 2.** (Continued)

| Core Based Statistical Area (CBSA) | Annual Miles Driven (mil.) | Percent National Miles Driven | Erlang B | Erlang C (delay) | Erlang C (mean wait) | Erlang C (excess wait) | Normal Approx. (z = 0.25) | Normal Approx. (z = 0.5) | Normal Approx. (z = 1) | Normal Approx. (z = 2) | Normal Approx. (z = 4) |
|---|---|---|---|---|---|---|---|---|---|---|---|
| Minneapolis-St. Paul-Bloomington | 34,540 | 1.6 | 46 | 48 | 59 | 59 | 38 | 41 | 45 | 51 | 58 |
| Nashville-Davidson-Murfreesboro–Franklin | 12,120 | 0.6 | 21 | 22 | 41 | 41 | 16 | 18 | 20 | 24 | 29 |
| New Orleans-Metairie | 5,528 | 0.3 | 13 | 14 | 35 | 35 | 9 | 10 | 12 | 15 | 18 |
| New York-Newark-Jersey City | 93,512 | 4.4 | 103 | 107 | 108 | 108 | 91 | 96 | 102 | 111 | 122 |
| Oklahoma City | 11,237 | 0.5 | 20 | 21 | 40 | 40 | 15 | 17 | 19 | 23 | 27 |
| Orlando-Kissimmee-Sanford | 16,728 | 0.8 | 27 | 28 | 44 | 44 | 20 | 23 | 26 | 30 | 35 |
| Philadelphia-Camden-Wilmington | 32,781 | 1.6 | 44 | 46 | 58 | 58 | 36 | 39 | 43 | 49 | 56 |
| Phoenix-Mesa-Scottsdale | 31,408 | 1.5 | 43 | 44 | 57 | 57 | 35 | 38 | 42 | 47 | 54 |
| Pittsburgh | 11,955 | 0.6 | 21 | 22 | 40 | 40 | 16 | 18 | 20 | 24 | 29 |
| Portland-Vancouver-Hillsboro | 17,096 | 0.8 | 27 | 28 | 45 | 45 | 21 | 23 | 26 | 30 | 36 |
| Providence-Warwick | 9,966 | 0.5 | 19 | 20 | 39 | 39 | 14 | 15 | 18 | 21 | 26 |
| Raleigh | 12,675 | 0.6 | 22 | 23 | 41 | 41 | 16 | 18 | 21 | 25 | 30 |
| Richmond | 10,501 | 0.5 | 19 | 20 | 39 | 39 | 14 | 16 | 18 | 22 | 26 |
| Riverside-San Bernardino-Ontario | 25,856 | 1.2 | 37 | 38 | 52 | 52 | 29 | 32 | 36 | 41 | 47 |
| Rochester | 6,792 | 0.3 | 15 | 15 | 36 | 36 | 10 | 12 | 14 | 17 | 20 |
| Sacramento–Roseville–Arden-Arcade | 16,946 | 0.8 | 27 | 28 | 45 | 45 | 21 | 23 | 26 | 30 | 36 |
| Salt Lake City | 6,616 | 0.3 | 15 | 15 | 36 | 36 | 10 | 11 | 14 | 16 | 20 |
| San Antonio-New Braunfels | 16,679 | 0.8 | 27 | 28 | 44 | 44 | 20 | 23 | 26 | 30 | 35 |
| San Diego-Carlsbad | 22,605 | 1.1 | 33 | 35 | 49 | 49 | 26 | 29 | 32 | 37 | 43 |
| San Francisco-Oakland-Hayward | 28,735 | 1.4 | 40 | 41 | 54 | 54 | 32 | 35 | 39 | 44 | 51 |
| San Jose-Sunnyvale-Santa Clara | 13,442 | 0.6 | 23 | 24 | 42 | 42 | 17 | 19 | 22 | 26 | 31 |
| Seattle-Tacoma-Bellevue | 17,773 | 0.8 | 28 | 29 | 45 | 45 | 22 | 24 | 27 | 31 | 37 |
| St. Louis | 19,770 | 0.9 | 30 | 31 | 47 | 47 | 23 | 26 | 29 | 34 | 40 |
| Tampa-St. Petersburg-Clearwater | 22,121 | 1.1 | 33 | 34 | 49 | 49 | 26 | 28 | 32 | 36 | 43 |
| Virginia Beach-Norfolk-Newport News | 8,893 | 0.4 | 18 | 18 | 38 | 38 | 12 | 14 | 16 | 20 | 24 |

*(Continued)*

**Table 2.** (Continued)

| Core Based Statistical Area (CBSA) | Annual Miles Driven (mil.) | Percent National Miles Driven | Erlang B | Erlang C (delay) | Erlang C (mean wait) | Erlang C (excess wait) | Normal Approx. ($z = 0.25$) | Normal Approx. ($z = 0.5$) | Normal Approx. ($z = 1$) | Normal Approx. ($z = 2$) | Normal Approx. ($z = 4$) |
|---|---|---|---|---|---|---|---|---|---|---|---|
| Washington-Arlington-Alexandria | 41,199 | 2.0 | 53 | 55 | 65 | 65 | 44 | 47 | 52 | 58 | 66 |

servers occur in our Gaussian approximation for different peakedness parameters. Thus, we see that if the variance of disengagement interarrival times is high, then a larger number of remote drivers is needed.

## Author Contributions

**Conceptualization:** Robert C. Hampshire, Shan Bao, Walter S. Lasecki, Andrew Daw, Jamol Pender.

**Data curation:** Robert C. Hampshire.

**Formal analysis:** Andrew Daw.

**Funding acquisition:** Robert C. Hampshire.

**Investigation:** Robert C. Hampshire, Jamol Pender.

**Methodology:** Robert C. Hampshire, Walter S. Lasecki, Andrew Daw, Jamol Pender.

**Project administration:** Robert C. Hampshire.

**Supervision:** Robert C. Hampshire, Jamol Pender.

**Writing – original draft:** Robert C. Hampshire, Andrew Daw, Jamol Pender.

**Writing – review & editing:** Robert C. Hampshire, Shan Bao, Walter S. Lasecki, Andrew Daw, Jamol Pender.

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
