## [Decision Letter · Decision Letter 0]

18 Mar 2020

PONE-D-20-04402

Beyond Safety Drivers: Applying air traffic control principles to support the deployment of driverless vehicles

PLOS ONE

Dear Prof. Hampshire,

Thank you for submitting your manuscript to PLOS ONE. After careful consideration, we feel that it has merit but does not fully meet PLOS ONE’s publication criteria as it currently stands. Therefore, we invite you to submit a revised version of the manuscript that addresses the points raised during the review process.

We would appreciate receiving your revised manuscript by May 02 2020 11:59PM. To enhance the reproducibility of your results, we recommend that if applicable you deposit your laboratory protocols in protocols.io, where a protocol can be assigned its own identifier (DOI) such that it can be cited independently in the future. For instructions see: http://journals.plos.org/plosone/s/submission-guidelines#loc-laboratory-protocols

We look forward to receiving your revised manuscript.

Kind regards,

Yanyong Guo, Ph.D

Academic Editor

PLOS ONE

Journal Requirements:

2. We note that Figure 1 in your submission contain copyrighted images. All PLOS content is published under the Creative Commons Attribution License (CC BY 4.0), which means that the manuscript, images, and Supporting Information files will be freely available online, and any third party is permitted to access, download, copy, distribute, and use these materials in any way, even commercially, with proper attribution. For more information, see our copyright guidelines: http://journals.plos.org/plosone/s/licenses-and-copyright.

1.         You may seek permission from the original copyright holder of Figure(s) [#] to publish the content specifically under the CC BY 4.0 license.

3. We note that Figure 1 includes an image of a participant in the study. 

Reviewers' comments:

Reviewer's Responses to Questions

**Comments to the Author**

1. Is the manuscript technically sound, and do the data support the conclusions?

Reviewer #1: Yes

Reviewer #2: Yes

2. Has the statistical analysis been performed appropriately and rigorously? 

Reviewer #1: Yes

Reviewer #2: N/A

3. Have the authors made all data underlying the findings in their manuscript fully available?

Reviewer #1: Yes

Reviewer #2: Yes

4. Is the manuscript presented in an intelligible fashion and written in standard English?

Reviewer #1: Yes

Reviewer #2: Yes

5. Review Comments to the Author

Reviewer #1: Very interesting paper and very good job. A nationwide remote monitoring system for driverless vehicles are proposed, which could increase safety dramatically, speed these vehicles’ deployment, and provide employment. The remote operators do not directly drive the vehicles, but provide input on high level tasks such as path-planning, object detection and classification. No more comments.

Reviewer #2: This is an interesting paper, it is well written and easy to follow. I have the following comments:

(1) on Page 8, the authors assumed arrival rate is constant. This assumption, in my opinion, is kind of over simple. If we look at the distribution of crashes (assuming that number of crashes and remote drivers needed are proportional), it is not evenly distributed by hour, day (week day vs weekend), as well as other conditions (roadway type, weather, etc.). Taking the fallen tree as an example, significant more remote drivers may be needed under adverse weathers. I encourage the authors discuss it in the paper.

(2) Lines 103 to 107 on Page 4, can you further clarify how you come up with the number of 6.25 m disengagements?

(3) Minor: first line on Page 10. “Inter-arrival and service distributions” should be “Inter-arrival and service time distributions?” missing “time”?

(4) One question not related to the work in this paper, but I am curious. How about if the remote driver makes mistakes and “causes” a crash? Should he/she take responsibility?

6. PLOS authors have the option to publish the peer review history of their article (what does this mean?). If published, this will include your full peer review and any attached files.

Reviewer #1: No

Reviewer #2: No

---

## [Author Response · Author response to Decision Letter 0]

25 Mar 2020

see attached response document.

---

## [Decision Letter · Decision Letter 1]

23 Apr 2020

Beyond Safety Drivers: Applying air traffic control principles to support the deployment of driverless vehicles

PONE-D-20-04402R1

Dear Dr. Hampshire,

We are pleased to inform you that your manuscript has been judged scientifically suitable for publication and will be formally accepted for publication once it complies with all outstanding technical requirements.

With kind regards,

Yanyong Guo, Ph.D

Academic Editor

PLOS ONE

Additional Editor Comments (optional):

Reviewers' comments:

Reviewer's Responses to Questions

**Comments to the Author**

1. If the authors have adequately addressed your comments raised in a previous round of review and you feel that this manuscript is now acceptable for publication, you may indicate that here to bypass the “Comments to the Author” section, enter your conflict of interest statement in the “Confidential to Editor” section, and submit your "Accept" recommendation.

Reviewer #1: All comments have been addressed

Reviewer #2: All comments have been addressed

2. Is the manuscript technically sound, and do the data support the conclusions?

Reviewer #1: Yes

Reviewer #2: Yes

3. Has the statistical analysis been performed appropriately and rigorously? 

Reviewer #1: Yes

Reviewer #2: Yes

4. Have the authors made all data underlying the findings in their manuscript fully available?

Reviewer #1: Yes

Reviewer #2: Yes

5. Is the manuscript presented in an intelligible fashion and written in standard English?

Reviewer #1: Yes

Reviewer #2: Yes

6. Review Comments to the Author

Reviewer #1: The authors have properly replied the my previous concerns. So I suggest that it could be published.

Reviewer #2: (No Response)

7. PLOS authors have the option to publish the peer review history of their article (what does this mean?). If published, this will include your full peer review and any attached files.

Reviewer #1: No

Reviewer #2: No

---

## [Editor Report · Acceptance letter]

8 May 2020

PONE-D-20-04402R1 

Beyond Safety Drivers: Applying air traffic control principles to support the deployment of driverless vehicles 

Dear Dr. Hampshire:

I am pleased to inform you that your manuscript has been deemed suitable for publication in PLOS ONE. Congratulations! Your manuscript is now with our production department. 

With kind regards,

on behalf of

Dr. Yanyong Guo 

Academic Editor

PLOS ONE